# A 6-Item Family Resilience Scale (FRS6) for Measuring Longitudinal Trajectory of Family Adjustment

**DOI:** 10.3390/bs14030251

**Published:** 2024-03-20

**Authors:** Catherine So Kum Tang, Tiffany Sok U Siu, Tak Sang Chow, Sin Hang Kwok

**Affiliations:** 1Department of Counselling and Psychology, Hong Kong Shue Yan University, Hong Kong; 2Mrs Dorothy Koo and Dr Ti Hua Koo Centre for Interdisciplinary Evidence-Based Practice and Research, Hong Kong Shue Yan University, Hong Kong

**Keywords:** 6-item FRS6, family resilience, life adversities, CPR model of family resilience

## Abstract

Two studies were conducted in Hong Kong to validate a brief measure of family resilience based on the three-factor CPR model. The CPR model stipulates that family resilience comprises three major factors: Communication and Connectedness (C), Positive Framing (P), and External Resources (R). Study 1 abbreviated the 16-item Family Resilience Scale (FRS16) into six items (FRS6) with a parent sample in the community (N = 1270). Study 2 tested the validity of the FRS6 with a single parent sample (N = 336). The result of Study 1 suggests a dominant general family resilience factor structure with three distinct subfactors. The highest factor loading items from each of the three subfactors were retained in the six-item FRS6. The resultant FRS6 was internally consistent and related to various correlates in similar or better strengths as compared to the FRS16. The results of a separate sample in Study 2 indicated that the FRS6 demonstrated satisfactory internal reliability and correlated significantly with individual resilience, quality of life, anxiety, and depression in the expected directions. Both studies supported that the FRS6 is a psychometrically sound measure of family resilience and can be used in longitudinal studies that aim to chart the trajectory of family adjustment following life adversities.

## 1. Introduction

Family resilience refers to a family’s capabilities to withstand, rebound, and thrive from adversities [1]. Based on Walsh’s framework [1,2], the CPR model stipulates that family resilience comprises three major factors: Communication and Connectedness (C), Positive Framing (P) and External Resources (R) [3]. Communication and Connectedness refers to a family’s ability to communicate openly in recognizing and solving problems as well as stay connected with each other for support. Positive Framing refers to a family’s ability to remain hopeful and make meaning out of adversities, while External Resources refers to external support (e.g., community network) available to the family when adversities emerge.

There is growing empirical support for the beneficial effects of family resilience. For example, family resilience was found to be associated with better mental health under the recent COVID-19 pandemic outbreak [4,5,6,7,8] as well as among parents of children with special needs [9] and medical disorders [10]. Aside from its protective role toward negative outcomes, a recent study has also found that family resilience was associated with flourishing among children who experienced different degrees of adversities such as adverse childhood experiences, disadvantaged family financial background, and special health care needs [11]. The benefits of family resilience have also been revealed in the practical realm. For instance, an intervention program that aimed at enhancing family resilience was shown to buffer the negative neighborhood effect on children’s delinquent behaviors [12].

Relatively few researchers have examined how family resilience may change over the course of life adversities and affects the adjustment of individuals and their families [13]. The successive waves of the COVID-19 pandemic over the past three years have disrupted the daily routines and mental health of individuals as well as their families [4,14]. This has provided researchers with an opportunity to assess family resilience and its impact during and after encountering adversities. To this end, a brief measurement of family resilience is important in order to chart the trajectory of family resilience before and after adversities. 

The present study aimed to abbreviate a family resilience scale and examine whether it demonstrates similar associations with psychosocial factors of individual resilience, depression, anxiety, and quality of life as compared to a longer version. The selection of items for the abbreviated scale would be determined both theoretically and statistically. We hypothesized that the brief family resilience scale would demonstrate similar associations to relevant variables to the longer versions of the family resilience scale [15,16,17]. Specifically, the abbreviated scale would be negatively associated with anxiety and depression but positively related to quality of life and individual resilience. 

## 2. Method

### 2.1. Procedures 

Two studies were conducted with two separate Hong Kong samples. Data were collected for both studies in May to July 2022 amidst the major COVID-19 outbreak dominated by the Omicron variant in Hong Kong [18]. For Study 1, parents were recruited from the general community via an online survey platform, whereas participants of Study 2 were recruited through a social service organization for single parents in Hong Kong. Both samples went through identical study procedures to complete identical online surveys. Firstly, they were presented with the purposes of the study; then, they indicated their understanding of rights and consent by pressing “continue to next page”. Participants answered the questions anonymously and were compensated with a supermarket coupon valued at HKD 50 (about USD 6) upon completion. Approval for these studies was granted by the Human Research Ethics Committee of the research team’s affiliated university. 

### 2.2. Measures

#### 2.2.1. Family Resilience

Family resilience was measured with the FRS16 [3]. This 16-item scale includes the CPR components of family resilience: Communication and Connectedness, Positive Framing, and Resources. The scale was previously validated with a Chinese sample in mainland China and an American sample. The Cronbach’s alpha for the overall scale was 0.78 for the mainland Chinese sample. Each of the three subfactors as well as the total score of the FRS16 were found to be associated with quality of life, general health, family cohesion, relationship satisfaction and perceived community support in Chinese [3]. Responses to the scale were made on a 4-point Likert scale (1 as *strongly disagree* to 4 as *strongly agree*). A sample item is “We consult with each other about decisions”. Scores are summed with higher scores indicating a higher level of family resilience.

#### 2.2.2. Individual Resilience

Individual resilience was measured with the Chinese version of the Connor–Davidson Resilience Scale [3,19] which consists of 10 items (e.g., “I can handle unpleasant feelings”). Participants rated their responses from 0 (*not true at all*) to 4 (*true nearly all the time*). For the present study, Cronbach’s alpha for samples 1 and 2 were 0.95 and 0.94, respectively.

#### 2.2.3. Quality of Life

The Chinese version of the EUROHIS-QOL [3,20] was used to measure participants’ level of satisfaction in different aspects of life (e.g., “How would you rate your quality of life?”). Responses were made on a 5 point-scale from 1 (*not at all*) to 5 (*completely*). For the present study, the Cronbach’s alpha was satisfactory across the two samples (0.91). 

#### 2.2.4. Anxiety and Depression

Levels of anxiety and depression were measured with the Chinese version of the ultra-brief screening scale for anxiety and depression (PHQ-4) [3,21]. Participants indicated how anxious and depressed they felt over the past two weeks (e.g., “Over the last 2 weeks, how often have you been feeling nervous, anxious, or on edge?”) on a 5-point scale (0 = *not at all*, 4 = *nearly every day*). For the present study, the internal reliability was satisfactory across the two samples. 

### 2.3. Data Analysis 

#### 2.3.1. Study 1: Scale Abbreviation and Validation

In Study 1, we abbreviated the FRS16 [3] with 1270 parents in Hong Kong based on the Exploratory Structural Equation Modeling analysis (ESEM). ESEM incorporates both elements of confirmatory (CFA) and exploratory factor analysis (EFA) in one analytic framework. ESEM constrains the cross-loadings on non-target factors to be as close to zero as possible [22]. Researchers have recommended using ESEM as it yields more reliable results compared to CFA [23]. Based on the 3-factor CPR model of family resilience [3], a priori assumption was made that family resilience consists of three factors: Communication and Connectedness (C), Positive Framing (P), and External Resources (R). 

We tested the following models: Model 1 (first-order three-factor ESEM) which posits that family resilience is comprised of three inter-related factors of CPR, Model 2 (higher-order three-factor ESEM) with which family resilience is construed as a second-order factor and is explained by three inter-related factor, and Model 3 (bifactor ESEM) stipulates that family resilience consists of a general factor of family resilience along with three distinct and independent subfactors. The decision of the best-fitting model was based on both the model fit indices and measurement quality [22]. We consider the following fit indices: Chi-square (χ^2^), Root-Mean-Square Error of Approximation (RMSEA), Standardized Root Mean Square Residual (SRMR), Comparative Fit Index (CFI), and Tucker–Lewis Index (TLI). Excellent model fits would demonstrate RMSESA and SRMR from 0.01 to 0.05 and 0.96 to 0.99 for both CFI and TLI [22]. 

All models were submitted to Mplus 8.3 [24], and codes for ESEM models were generated with the help of ESEM Code Generator [25]. Family resilience is typically measured as a continuous variable, the ESEM models were then specified with an ML estimator and estimated with the target rotation method [22]. Item selections for the abbreviated version will be based on these criteria: (1) item–total correlation and magnitude of standardized factor loading coefficients, wherein the larger coefficient the better; (2) items need to be internally consistent; (3) breadth content of original scale (FRS16) should be retained; and (4) face validity, wherein items purportedly measure family resilience.

We also examined whether the abbreviated family resilience scale performed similarly to the FRS16 by correlating it with relevant constructs of individual resilience, anxiety and depression, and quality of life. The internal reliability of the abbreviated scale was also examined. 

#### 2.3.2. Study 2: Cross Sample Validation 

The abbreviated family resilience scale in Study 1 was validated with a separate sample to determine whether it performed similarly to FRS16 among single parents in Hong Kong. We also examined whether the brief scale correlated with relevant constructs of individual resilience, anxiety and depression, and quality of life in a similar magnitude and direction as FRS16. The internal reliability of the brief scale was also examined. 

## 3. Results

### 3.1. Study 1: Abbreviation and Model Testing 

Table 1 shows the demographic characteristics of the final group of participants (N = 1270) in Study I. Participants were fairly gender balanced. The majority of them were married, highly educated, had one child or two children, and were employed either full- or part-time. 

As shown in Table 2, only the bifactor ESEM model (Model 3) had all the fit indices meeting conventional cut-offs. However, the parameter estimates of this model show that item 13 (“We attend church/synagogue/mosque services”) had a standardized loading larger than 1 on the Resources factor; thus, R^2^ could not be identified. Considering we have a relatively large sample size, the problem might be that item 13 was performing differently than the other items; therefore, it could not be grouped to the specific factor. In other studies that validated family resilience scales in China, the spirituality factor was also dropped due to low item-to-scale correlations [26] or low factor loadings [27]. It is possible that a cultural difference exists in people’s understanding of the relationship between family resilience and spirituality. Therefore, we modified the bifactor model with the removal of item 13, and the resultant final modified bifactor model showed an excellent fit (CFA = 0.98; TLI = 0.97; RMSEA = 0.04; SRMR = 0.02). 

Table 3 shows that each item was adequately associated with the overall family resilience score with all corrected item–total correlation values larger than 0.3 except for item 14. The scale also has a well-defined general factor in which all items significantly loaded on the general factor (λ > 0.35 and small standard errors < 0.04) with item 14 being the only exception. In addition, the general factor emerges as the most dominant factor in the modified bifactor model, explaining 54% of the common variance. To cover the content breadth of the factors, we retained items with the four highest loadings items from Communication and Connectedness (C: Items 1, 2, 4, and 5) and the single highest loading item each from Positive Framing (P: Item 10) and External Resources (R: Item 7). In other words, the brief family resilience scale (hereafter referred to as FRS6) includes six items from the FRS16 (see Table 3 for the full scale).

The FRS6 was found to be internally consistent (alpha = 0.79) and correlated with FRS16 (r = 0.91, *p* < 0.001). Both FRS16 and FRS6 correlated positively to individual resilience and quality of life but related negatively to anxiety and depression (Table 4).

### 3.2. Study 2: Cross Sample Validation of FRS6 

The second sample consisted of 336 single parents recruited from a social service organization in Hong Kong. Sample characteristics are also summarized in Table 1. This group of participants was mainly females (94.3%), 64.3% completed secondary school, and 74.1% were divorced, separated, or widowed. Most of them had only one child (60.1%), were unemployed (60.5%), and had a yearly household income of US$1911 or less (81%). Study 2 provided a cross-sample validation of FRS6 with participants from more diverse family characteristics than the community parent sample in Study 1.

As shown in Table 4, FRS6 was highly correlated with FRS16 (r = 0.92, *p* < 0.005). Additionally, FRS6 showed satisfactory internal reliabilities (alpha = 0.80), although sizes were smaller compared to FRS16 as the former has a reduced number of items. FRS6 performed similarly or even better in their correlations with anxiety and depression, individual resilience, and quality of life in the hypothesized directions.

## 4. Discussion

A short and easily administered measurement of family resilience that tracks the trajectory of family functioning is lacking [13]. We abbreviated the FRS16 and tested the reliability and validity of the shortened family resilience scale (FRS6) with two separate samples in Hong Kong. Results indicated that the FRS16 consists of a general factor with three subfactors as stipulated in the CPR model [3]. We selected the six highest factor loading items that cover all three components of the CPR model to form the FRS6. The resultant six-item FRS6 demonstrated a comparable level of internal reliability as compared to the FRS16 despite a significant reduction in the number of scale items. In general, the FRS6 performed similarly or even better than the FRS16 in terms of its correlations with anxiety and depression, quality of life, and individual resilience across the two samples.

In past studies, the structural validations of family resilience scales yielded inconsistent results. The number of factors of family resilience varies from three to seven across different cultures and populations from the general public to individuals with specific adversities [28]. This might be attributed to the use of the overly restrictive CFA in previous studies. More recently, researchers have found that the ESEM modeling analysis yielded more reliable results as compared to CFA. They have argued that the ESEM approach has the potential to address cultural differences in interpreting the same construct [13]. The present study is the first to utilize the less restrictive ESEM approach to examine the underlying structure of family resilience with Hong Kong samples. 

Our results supported the three-factor CPR structure of family resilience of previous studies [3]. We found the FRS6 demonstrating satisfactory construct validity across two samples with different family characteristics in Hong Kong. The FRS6 also compared favorably with the FRS16. In particular, the FRS6 was correlated with individual resilience, anxiety and depression, and quality of life in similar or even better strengths as compared to the FRS16 for both Hong Kong samples. This provides support that the FRS6 is a valid alternative to the longer versions of family resilience scales. According to the results in Study 1, the modified bifactor model has the best fit model indices. This suggests a general factor with three subfactor structures best fitted the family resilience construct. For this reason, we recommend using a single total score of the FRS6 to measure family resilience as in recent empirical studies [3,7,28]. The FRS6 should be scored on a 4-point Likert scale (1 = *strongly disagree* to 4 = *strongly agree*) to ensure consistency and accuracy. When measuring family resilience over time, the time period of FRS6 should also be specified (e.g., in the last six months, etc.).

The development and validation of the FRS6 contributes significantly to a more detailed understanding of family resilience. According to theoretical frameworks of family resilience [1,2,3], family resilience is conceptualized as a dynamic and ongoing process rather than a static trait. Family resilience is not something that individuals or families possess or lack but rather something that is developed and nurtured over time. By examining family resilience before, during and after adversities, researchers can gain a deeper understanding of how the resilience processes unfold in real-life situations. However, there is a lack of longitudinal studies that observe and measure family resilience over time, particularly in the context of adversities [13]. Additionally, researchers have advocated to examine family resilience in an ecological framework in which family resilience is part of the multiple systems of an individual [8,12,29]. Most empirical research has been stagnant in the individual scope. The development of the FRS6 allows researchers to track changes of family resilience over time and subsequent to different life crises while reserving space for assessing antecedent and criterion variables beyond individual factors. 

The FRS6 stands out for its brevity and efficiency compared to other existing measures. The existing measures in family resilience consist of a large number of items ranging from 16 to 54 items [2,3], which can be time-consuming and burdensome for participants. The FRS6 only includes six items and thus can reduce participant fatigue and allows for quicker data collection, making it more feasible for use in research or clinical settings where time is limited. The brevity and simplicity of the FRS6 also make it practical and user-friendly. It can be easily administered and scored, its user-friendly nature enhances its utility and allows the inclusion of more participants, especially elderly individuals who may have difficulty in reading [29]. Despite its brevity, the FRS6 captures key aspects of family resilience, including family communication and connectedness, positive mindset, and the availability of external resources. By focusing on these essential dimensions, the scale provides a concise assessment of family resilience. 

The present study has strengths including validations with two independent samples and the incorporation of a more powerful statistical method of ESEM. However, it also comes with limitations. Firstly, we recruited two separate samples, and most participants were parents. In addition, the cross-validation sample included more single mothers than single fathers. As such, the generalizability of the FRS6 requires stronger empirical support from more representative samples. Future studies should test whether the FRS6 can be generalized to samples with more diverse characteristics such as single fathers, elders and young children. Secondly, examination of test–retest reliability is not available in the current study; therefore, the stability of the FRS6 is yet to be verified. As mentioned, family resilience is often considered as a process [1,30] with fluctuation expected alongside the occurrence of crucial life events and challenges. This notion has also been supported partially, as family resilience measured in two time points only showed moderate correlations [8]. Thus, is it expected that the FRS6 will perform in a similar manner. Our studies relied on the self-reports of one member of the family. Future studies should attempt to include the perceptions and experiences of other family members. 

Cultural norms, beliefs, and values play a significant role in shaping resilience and coping strategies within families. The FRS6 was validated with two Chinese samples, and this may involve modifying the scale items to reflect culturally specific experiences when the scale is used in different cultures. Furthermore, socioeconomic factors, such as income, education, and access to resources, can significantly influence family resilience. Families from different socioeconomic backgrounds may face unique stressors and have different levels of support available to them. It is important to consider these factors when interpreting the scale scores and to account for potential disparities in resilience levels across socioeconomic groups. Families come in various forms, and each structure may present distinct challenges and dynamics. For example, single-parent families may face different stressors compared to nuclear families, and extended families may have additional sources of support. Researchers and practitioners should consider how these structures may impact the measurement of family resilience.

## 5. Practical and Research Implications

The FRS6 is a useful tool for researchers, clinicians, and program evaluators to assess the strengths and weaknesses of families in different situations. In clinical settings, the FRS6 can be used to evaluate the resilience levels of families undergoing psychological intervention or facing specific life crisis and trauma. By assessing the family’s ability to adapt and rebound from these challenges, clinicians can gain a better understanding of the family’s overall well-being and tailor interventions to meet their specific needs. The FRS6 can also serve as an outcome measure to evaluate the effectiveness of interventions aimed at enhancing family resilience in clinical settings. In program evaluations, the FRS6 can be employed to assess the impact of resilience-building programs or interventions on families. For example, in community-based programs targeting at-risk groups, the FRS6 can be used to measure changes in resilience levels before and after program participation. This information can inform about the effectiveness of the program in improving family resilience and guide future improvements.

Future research directions can expand the application of the FRS6 to diverse populations and settings. It is important to ensure the validity and reliability of the FRS6 across different cultural, socioeconomic, and demographic groups. This may involve conducting studies with larger and more diverse samples to establish the generalizability of the scale’s psychometric properties. Further research is also needed to examine the sensitivity of the FRS6 to change over time and its ability to capture different aspects of family resilience. This involves longitudinal studies to assess how family resilience fluctuates in response to various stressors and interventions. Moreover, exploring the relationship of the FRS6 with other relevant constructs, such as family cohesiveness and adaptability, can provide a more comprehensive understanding of its implications and potential applications. Furthermore, future research should also examine the predictive validity of the scale in different contexts. For example, studying its ability to predict family outcomes, such as mental health, relationship satisfaction, or parenting behaviors, can provide valuable insights into the practical implications of measuring family resilience.

## 6. Conclusions

The recent global COVID-19 pandemic has called for more attention on cultivating and utilizing family resilience to cope with unexpected life adversities. A brief assessment tool for measuring the resilient trajectories of families is in high demand. We have devised a six-item FRS6 based on the CPR model of family resilience. The resultant FRS6 shows comparable internal reliability and associations with anxiety and depression, individual resilience, and quality of life in the expected direction with similar magnitude as using longer versions of family resilience scales. In sum, the FRS6 is a viable brief measurement for tracking trajectories of family resilience over time and in different situations.

## Figures and Tables

**Table 1 behavsci-14-00251-t001:** Sample characteristics for studies 1 and 2.

		Study 1 * Community Parents	Study 2 Single Parents
		**Frequency (%)**	**Frequency (%)**
Gender		N = 1270	N = 336
	female	621 (48.9%)	317 (94.3%)
	male	649 (51.1%)	19 (5.7%)
Education level			
	primary school	28 (2.2%)	61 (18.2%)
	secondary school	307 (24.2%)	216 (64.3%)
	tertiary education or above	784 (61.7%)	59 (17.6%)
Marriage status			
	single	41 (3.2%)	37 (11%)
	married	959 (75.5%)	50 (14.9%)
	divorced/separate	90 (7.1%)	200 (59.5%)
	widowed	29 (2.3%)	49 (14.6%)
No. of children			
	1	545 (42.9%)	202 (60.1%)
	2	488 (38.4%)	103 (30.7%)
	3 or more	86 (6.8%)	31 (0.09%)
Employment status			
	full-time	675 (53.1%)	48 (14.3%)
	part-time	92 (7.2%)	66 (19.6%)
	full-time homemakers	141 (11.1%)	182 (54.2%)
	unemployed/retired	205 (16.1%)	21 (6.3%)
	others	6 (0.5%)	19 (5.7%)
Monthly household income (in USD)			
	1019 or less	54 (4.3%)	134 (39.9%)
	1019–1911	68 (5.4%)	138 (41.1%)
	1911–2548	82 (6.5%)	42 (12.5%)
	2548–3822	154 (12.1%)	20 (6%)
	3822–5096	185 (14.6%)	0
	5096 or above	576 (45.4%)	2 (0.6%)
Age	M (SD)	48.13 (12.30)	44.40 (7.92)
	Range	18–78	24–65

Note: * Only 1119 participants in Study 1 completed information on education, marriage status, number of children, employment status, and monthly household income.

**Table 2 behavsci-14-00251-t002:** Model fit indices of ESEM models of Study 1 (N = 1270).

Model	Type	x^2^	df	x^2^/df	CFI	TLI	RMSEA		SRMR	AIC	BIC	aBIC	Meets Criteria
Model 1	First-order three factor ESEM	914.59	75	11.47	0.91	0.86	0.09	[0.09, 0.10]	0.05	27,719.51	28,115.81	27,871.22	Partially
Model 2	Second-order three factor ESEM	918.83	78	11.77	0.91	0.86	0.09	[0.09, 0.10]	0.05	27,717.74	28,098.60	27,863.55	Partially
Model 3	Bifactor ESEM	314.72	62	5.08	0.97	0.95	0.06	[0.05, 0.06]	0.02	27,145.63	27,608.84	27,322.96	Yes
**Final Model**	**Modified Bifactor ESEM ***	**173.78**	**51**	**3.41**	**0.98**	**0.97**	**0.04**	**[0.04, 0.05]**	**0.02**	**25,332.12**	**25,764.45**	**25,497.63**	**Yes**

Note: * Item 13 (“We attend church/synagogue/mosque services”) of the FRS16 was removed.

**Table 3 behavsci-14-00251-t003:** Item-level descriptive statistics and factor loadings of final modified bifactor ESEM model in Study 1.

Factor	Item	Mean	SD	CITC	General Factor			C Subfactor		P Subfactor		R Subfactor	
**Communication and Connectedness (C)**					**λ**	**S.E.**	**R^2^**	**λ**	**S.E.**	**λ**	**S.E.**	**λ**	**S.E.**
FRS_1 *	We can compromise when problems come up.	2.81	0.56	0.52	**0.51**	0.02	0.32	**0.25**	0.03	−0.04	0.03	0.01	0.03
FRS_2 *	We can talk about the way we communicate in our family.	2.92	0.51	0.62	**0.51**	0.02	0.57	**0.56**	0.02	0.04	0.03	−0.04	0.02
FRS_3	We consult with each other about decisions.	2.96	0.52	0.61	**0.48**	0.03	0.56	**0.58**	0.02	0.04	0.03	−0.01	0.02
FRS_4 *	We define problems positively to solve them.	2.96	0.50	0.65	**0.54**	0.02	0.64	**0.59**	0.02	0.00	0.03	0.00	0.02
FRS_5 *	We discuss problems and feel good about the solutions.	2.85	0.54	0.68	**0.54**	0.02	0.71	**0.65**	0.02	−0.01	0.02	0.01	0.02
FRS_6	We discuss things until we reach a resolution.	2.81	0.57	0.62	**0.49**	0.03	0.58	**0.57**	0.02	** −0.11 **	0.03	** 0.07 **	0.02
FRS_11	We will not be taken for granted by family members.	2.65	0.62	0.41	**0.37**	0.03	0.18	**0.19**	0.03	0.04	0.04	** 0.07 **	0.03
FRS_12	We often listen to family members concerns or problems.	2.87	0.55	0.54	**0.45**	0.03	0.36	**0.39**	0.03	0.05	0.03	0.02	0.02
**Positive Framing (P)**													
FRS_9	We can solve major problems.	2.73	0.61	0.56	**0.72**	0.04	0.60	0.01	0.03	**−0.28**	0.06	0.01	0.02
FRS_10 *	We can survive if another problem comes up.	2.85	0.53	0.61	**0.83**	0.03	0.78	−0.02	0.03	**−0.29**	0.09	** −0.07 **	0.02
FRS_15	We accept stressful events as a part of life.	3.01	0.51	0.40	**0.55**	0.03	0.44	** −0.06 **	0.02	**0.36**	0.09	−0.03	0.02
FRS_16	We accept that problems occur unexpectedly.	2.87	0.52	0.51	**0.62**	0.03	0.46	0.02	0.03	**0.28**	0.06	0.01	0.02
**External Resources (R)**													
FRS_7 *	We feel people in this community are willing to help in an emergency.	2.61	0.68	0.48	**0.41**	0.03	0.79	** 0.04 **	0.02	0.01	0.02	**0.79**	0.04
FRS_8	We know there is community help if there is trouble.	2.45	0.69	0.48	**0.41**	0.03	0.77	0.02	0.02	−0.02	0.02	**0.77**	0.04
FRS_14	We participate in church activities.	2.02	0.77	0.24	**0.21**	0.03	0.10	0.02	0.03	0.04	0.04	**0.24**	0.03
Proportion of explained common variance (ECV)					0.54			0.25		0.05		0.16	

Note: * Items of the 6-item FRS6; bold items, significant target loadings (*p* < 0.05); underlined items indicate cross-loading items; S.E., standard error; CICT, corrected item total correlation.

**Table 4 behavsci-14-00251-t004:** Comparison of FRS6 and FRS16 with relevant psychological variables and estimates of reliability of two Hong Kong samples.

		Study I		Study 2	
		(N = 1270)		(N = 336)	
		FRS6	FRS16	FRS6	FRS16
1	Anxiety and depression	−0.319 ***	−0.298 ***	−0.155 ***	−0.104
2	Individual resilience	0.436 ***	0.449 ***	0.299 ***	0.323 ***
3	Quality of life	0.488 ***	0.490 ***	0.341 ***	0.347 ***
	Reliability	0.79	0.87	0.80	0.89
	Correlation between FRS6 and FRS16	0.911 ***		0.917 ***	

Note: *** *p* < 0.001.

## Data Availability

The data supporting the conclusions of this article will be made available by the authors with reasonable request.

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
