# Peer review of "A 6-Item Family Resilience Scale (FRS6) for Measuring Longitudinal Trajectory of Family Adjustment"

_behavsci, 2024, doi:10.3390/bs14030251_

Round 1
Reviewer 1 Report
Comments and Suggestions for Authors
Review of “A 6-item Family Resilience Scale (FRS6) for Measuring Longitudinal Trajectory of Family Adjustment" - behavsci-2892735-peer-review-v1
The study abbreviated the 16-item Family Resilience Scale (FRS16) into six items (FRS6), and is based on Walsh’s conceptualization that family resilience is composed of three major factors of communication and connectedness, positive framing, and external resources.
The importance of this study and of building an abbreviated tool by which to measure family resilience is uncontested; however, the introduction should elaborate some more on the factors comprising family resilience according to Froma Walsh.
Method
Two studies were conducted with two separate samples amidst the COVID-19 and Omicron variant outbreaks in Hong Kong.
To abbreviate the 16 items of FRS16 and validate the new measure, measures examining depression, anxiety, quality of life, and individual resilience were employed , as used in the longer version.
Results
The author/s did their best to abbreviate the 16 items of FRS16 with two large samples of parents and single-parent families.
Discussion
I suggest that the author/s elaborate on the essence of family resilience rather than make assumptions such as: "In the theorization of family resilience, it has been defined as a process. However, in reality, it is seldom operationalized as a process in empirical research with longitudinal observation of family resilience before, during, and after adversity".
Author Response
- “I suggest that the author/s elaborate on the essence of family resilience rather than make assumptions such as: "In the theorization of family resilience, it has been defined as a process. However, in reality, it is seldom operationalized as a process in empirical research with longitudinal observation of family resilience before, during, and after adversity".
Reply: (line 235-242)
According to theoretical frameworks of family resilience, family resilience is conceptualized as a dynamic and ongoing process rather than a static trait. Family resilience is not something that individuals or families possess or lack, but rather something that is developed and nurtured over time. By examining family resilience before, during and after adversities, researchers can gain a deeper understanding of how the resilience processes unfold in real-life situations. However, there is a lack of longitudinal studies that observe and measure family resilience over time, particularly in the context of adversities.
Reviewer 2 Report
Comments and Suggestions for Authors
I have reviewed the manuscript "A 6-item Family Resilience Scale (FRS6) for Measuring Longitudinal Trajectory of Family Adjustment" (behavsci-2892735) and found the study well-conducted and relevant. The manuscript contributes significantly to the field of family resilience research by offering a validated, efficient, and user-friendly tool for assessing family resilience. However, some minor revisions are required to improve the quality and clarity of the manuscript.
The authors should consider the following recommendations:
1. The manuscript would benefit from minor improvements in language and clarity. Careful proofreading and editing are needed to improve overall readability.
2. The authors might consider a more detailed discussion on the potential implications of the FRS6 scale in various contexts, such as clinical settings or program evaluations. Additionally, while the validation process is thorough, future research directions could be expanded, especially in diverse populations and settings.
3. While the paper mentions the application of FRS6 in different populations, discussing any limitations or considerations when using the scale across diverse cultural, socioeconomic, and family structures would be beneficial.
4. Expand the discussion to compare FRS6 with other existing measures of family resilience or similar constructs. Highlighting the unique contribution and advantages of FRS6 would provide a more precise justification for its use.
5. Address any technical clarifications, such as the scoring system, interpretation of scores, and any precautions or recommendations for administering the scale to ensure consistency and accuracy.
Comments on the Quality of English LanguageThe manuscript would benefit from minor improvements in language and clarity. Careful proofreading and editing are needed to improve overall readability.
Author Response
Reviewer 2’s comment
- “The manuscript would benefit from minor improvements in language and clarity. Careful proofreading and editing are needed to improve overall readability."
Reply:
The entire manuscript has been carefully proofread and edited to improve overall readability.
- The authors might consider a more detailed discussion on the potential implications of the FRS6 scale in various contexts, such as clinical settings or program evaluations. Additionally, while the validation process is thorough, future research directions could be expanded, especially in diverse populations and settings
Reply:
- Practical implications: (line 290-302)
The FRS6 is a useful tool for researchers, clinicians, and program evaluators to assess the strengths and weaknesses of families in different situations. In clinical settings, FRS6 can be used to evaluate the resilience levels of families undergoing therapy or facing specific life crisis and trauma. By assessing the family's ability to adapt and rebound from these challenges, clinicians can gain a better understanding of the family's overall well-being and tailor interventions to meet their specific needs. The FRS6 can also serve as an outcome measure to evaluate the effectiveness of interventions aimed at enhancing family resilience in clinical settings. In program evaluations, the FRS6 can be employed to assess the impact of resilience-building programs or interventions on families. For example, in community-based programs targeting at-risk families or those experiencing high levels of stress, the scale can be used to measure changes in resilience levels before and after program participation. This information can inform program developers and evaluators about the effectiveness of the program in improving family resilience and guide future improvements.
- Future research directions: (line 303-316)
Future research directions could expand the application of FRS6 to diverse populations and settings. It is important to ensure the validity and reliability of the FRS6 across different cultural, socioeconomic, and demographic groups. This would involve conducting studies with larger and more diverse samples to establish the generalizability of the scale's psychometric properties. Further research is also needed to examine the sensitivity of the FRS6 to change over time and its ability to capture different aspects of family resilience. This could involve longitudinal studies to assess how family resilience fluctuates in response to various stressors and interventions. Moreover, exploring the scale's relationship with other relevant constructs, such as individual resilience or family functioning, could provide a more comprehensive understanding of its implications and potential applications. Furthermore, it would be beneficial to investigate the predictive validity of the scale in different contexts. For example, studying its ability to predict family outcomes, such as mental health, relationship satisfaction, or parenting behaviors, can provide valuable insights into the practical implications of measuring family resilience.
- “While the paper mentions the application of FRS6 in different populations, discussing any limitations or considerations when using the scale across diverse cultural, socioeconomic, and family structures would be beneficial.”
Reply: (line 275-288)
Cultural norms, beliefs, and values play a significant role in shaping resilience and coping strategies within families. The FRS6 was validated with two Chinese samples, and this may involve modifying the scale items to reflect culturally specific experiences when the scale is used in different cultures. Furthermore, socioeconomic factors, such as income, education, and access to resources, can significantly influence family resilience. Families from different socioeconomic backgrounds may face unique stressors and have different levels of support available to them. It is important to consider these factors when interpreting the scale scores and to account for potential disparities in resilience levels across socioeconomic groups. Families come in various forms, and each structure may present distinct challenges and dynamics. For example, single-parent families may face different stressors compared to nuclear families, and extended families may have additional sources of support. The scale should be applicable and relevant to different family structures, and researchers and practitioners should consider how these structures may impact the measurement of family resilience.
- “Expand the discussion to compare FRS6 with other existing measures of family resilience or similar constructs. Highlighting the unique contribution and advantages of FRS6 would provide a more precise justification for its use.”
Reply: (line 248-259)
The FRS6 stands out for its brevity and efficiency compared to other existing measures. The existing measures in family resilience consist of a large number of items ranging from 16 to 54 items, which can be time-consuming and burdensome for participants. The FRS6 only includes 6 items and thus can reduce participant fatigue and allows for quicker data collection, making it more feasible for use in research or clinical settings where time is limited. The brevity and simplicity of the FRS6 also make it practical and user-friendly. It can be easily administered and scored, its user-friendly nature enhances its utility and allows inclusion of more participants, especially elderly individuals who may have difficulty in reading. Despite its brevity, the FRS6 captures all three key aspects of family resilience, including family communication and connectedness, positive mindset, and availability of external resources. By focusing on these essential dimensions, the FRS6 provides a concise yet comprehensive assessment of family resilience.
- “Address any technical clarifications, such as the scoring system, interpretation of scores, and any precautions or recommendations for administering the scale to ensure consistency and accuracy.”
Reply: (line 229-233)
We recommend using FRS6 as a single total score to measure family resilience as in recent empirical studies. The FRS6 should be scored on a 4-point Likert Scale (1= strongly disagree to 4= strongly agree) to ensure consistency and accuracy. When measuring family resilience over time, the time period of FRS6 should also be specified (e.g., in the last six months etc.)
Reviewer 3 Report
Comments and Suggestions for Authors
The research provide two studies conducted in Hong Kong to validate a brief measure of family resilience based on the 3-factor CPR model. The authors aimed to abbreviate a family resilience scale and examine whether it demonstrates similar associations with psychosocial factors of individual resilience, depression, anxiety, and quality of life as compared to a longer version.
The originality of this paper comes from the scope of the measurment. Given that most empirical research has been stagnant in the individual scope, the FRS6 assess antecedent and criterion variables beyond individual factors.
The study adds a brief measurement of family resilience in order to chart the trajectory of family resilience, demonstrating similar associations to relevant variables as the longer versions of the family resilience scale.
Methodology is well established.
The inequality of men and women in the 2nd sample should be included as a limitation. The conclusions are consistent with the evidence and arguments presented.
The references are appropriate.
Tables and figures are clear.
Author Response
- “The inequality of men and women in the 2nd sample should be included as a limitation. The conclusions are consistent with the evidence and arguments presented.”
Reply: (line 262-266)
Firstly, we recruited two separate samples, and most participants were parents. In addition, the cross-validation sample included more single mothers than single fathers. The generalizability of FRS6 requires stronger empirical support from more representative samples. Future studies should test whether FRS6 can be generalized to samples with more diverse characteristics such as single fathers, elders and young children.